# Semi-supervised Semantic Segmentation using Auxiliary Network

## Abstract

Recently, the convolutional neural networks (CNNs) have shown great success on semantic segmentation task. However, for practical applications such as autonomous driving, the popular supervised learning method faces two challenges: the demand of low computational complexity and the need of huge training dataset accompanied by ground truth. Our focus in this paper is semi-supervised learning for semantic segmentation. We wish to use both labeled and unlabeled data in the training process. A highly efficient semantic segmentation network is our platform, which achieves high segmentation accuracy with a low model size and high inference speed. We propose a semi-supervised learning scheme to improve segmentation accuracy by including extra images without labels. While most existing semi-supervised learning methods are designed based on the adversarial learning techniques, we present a new and different approach, which trains an auxiliary CNN network that validates labels (ground-truth) on the unlabeled images. In the supervised training phase, both the segmentation network and the auxiliary network are trained using the labeled images. Then, in the unsupervised training phase, the unlabeled images are segmented and a subset of image pixels are picked up by the auxiliary network; and then they are used as ground truth to train the segmentation network. Thus, at the end, all dataset images can be used for retraining the segmentation network to improve the segmentation results. We use Cityscapes and CamVid datasets to verify the effectiveness of our semi-supervised scheme. Our experimental results show that it can improve the mean IoU for about 1.2% to 2.9% on the challenging Cityscapes dataset.

## 1 Introduction

Semantic Segmentation, which identifies the category label of each pixel, is an important task in computer vision. A number of convolutional neural network (CNN) based semantic segmentation systems have been developed in recent years such as Chen et al. (2017a;b; 2018); Long et al. (2015); Zhao et al. (2017); Ronneberger et al. (2015). For practical applications such as autonomous driving, there is a high demand for real-time processing and sufficient training data. Hence, different research directions have been explored. For example, some researchers proposed various efficient network structures (Chen et al., 2019; Paszke et al., 2016; Poudel et al., 2018; Lo et al., 2018; Poudel et al., 2019; Yu et al., 2018; Zhao et al., 2018; Romera et al., 2017; Li et al., 2019), and the others focus on the weakly- or semi-supervised learning schemes (Hung et al., 2018; Lin et al., 2016; Bearman et al., 2016; Qi et al., 2016; Rajchl et al., 2016; Pathak et al., 2015; Papandreou et al., 2015; Hong et al., 2015; Luc et al., 2016; Dai et al., 2015; Pathak et al., 2014; Zhu et al., 2019; Mittal et al., 2019).

In this study, we first design an efficient segmentation network inherited from Chen et al. (2019) as our baseline model, and then propose a semi-supervised learning scheme. There are three training stages in our system. In the first stage, the segmentation network is trained using the labeled images. Then, in the second stage, an auxiliary network is trained using the segmentation network model trained in the previous stage and the labeled data to generate the confidence map. Inspired by Hung et al. (2018), we use the concept of confidence map to assign a confidence score to each segmented pixel on the unlabeled images. In their work (Hung et al., 2018), the confidence map is the output of a GAN discriminator network (Goodfellow et al., 2014), where the discriminator network learns to distinguish between the segmentation map and the ground truth map. However, we find that

the trusted (high confidence) regions in their confidence map are mostly located on the large target objects, and thus the effectiveness of semi-supervised learning is limited. To obtain more reliable small object labels, we adopt a different approach in generating the confidence map. In our approach, the confidence map is generated by an auxiliary (CNN) network, which is trained using the proposed auxiliary loss function. We carefully design the auxiliary loss function such that it leads to a reliable confidence map, particularly, on the small objects. In the third stage, the unlabeled images are used as inputs to the proposed system to generate the labels. Therefore, we can use both originally labeled and unlabeled images to retrain the segmentation network model to achieve a better performance in the end.

In summary, our main contributions of this work are as follows. First, Based on DSNet-fast (Chen et al., 2019), we design a powerful segmentation network, which achieves a very good balance between speed and accuracy. It produces 73.9% mean IoU on the Cityscapes testing set with a speed of 73 FPS on a single GTX 1080Ti card. Second, We propose a semi-supervised learning scheme with the notion of *auxiliary network*, which can be used to annotate the unlabeled images. Third, with the help of auxiliary network, our semi-supervised learning method can include the unlabeled images in training, which improves the segmentation accuracy.

## 2 RELATED WORK

Recently, CNNs have been widely used in many fields of computer vision. For the semantic segmentation task, FCN (Long et al., 2015) is a pioneer. It replaces the fully-connected layers of the classification network (Krizhevsky et al., 2012) with the convolution layers, and thus it can generate dense labels on the input image of the same size. Then, SegNet (Badrinarayanan et al., 2017) was subsequently proposed, which uses a symmetric encoder-decoder architecture for feature map down-sampling and up-sampling. The U-Net (Ronneberger et al., 2015) introduces the concatenation operation to up-sample the features with different levels. PSPNet (Zhao et al., 2017) and DeepLab (Chen et al., 2017a;b; 2018) propose the atrous spatial pyramid pooling (ASPP) module to integrate multi-scale features. There are many other studies on improving the segmentation results; however, for practical applications, a high inference speed network is very desirable. For example, ENet (Paszke et al., 2016) is a real-time segmentation model with good segmentation results. ICNet (Zhao et al., 2018) and BiSeNet (Yu et al., 2018) were recently proposed, and they aim at a better balance between speed and accuracy.

Another challenge in training a semantic segmentation network is the need of a large amount of labeled data (ground truth). The labeling cost of pixel-level annotation is extremely expensive. Hence, in recent years, the study of the weakly- and semi-supervised learning approaches to tackle this problem attracted a lot of attention. For the weakly supervised methods, the pixel-level annotation is no longer used to train the segmentation network, but instead, other forms of annotations obtained at low costs are used for training. Several different types of weakly-supervision have been studied, including bounding box supervision (Rajchl et al., 2016; Dai et al., 2015), scribbles supervision (Lin et al., 2016), point supervision (Bearman et al., 2016), and image-level supervision (Qi et al., 2016; Pathak et al., 2014). Particularly, the development of image-level supervision is the most popular one. Pathak et al. (2015) convert image-level labels to restrict the distribution of CNN output. Papandreou et al. (2015) combine image-level labels using EM algorithms to train the segmentation model. Wei et al. (2018) propose a generic classification network, which adopts the convolutional blocks with different dilated rates to generate dense object localization maps. It can produce reliable segmentation masks for training the segmentation model. For semi-supervised learning, Hong et al. (2015) propose a method to separately train the classification and segmentation networks, and then pass the information from the classification network to the segmentation network to reduce the search space for effective segmentation. Luc et al. (2016) employ an adversarial network to enhance the segmentation quality. Recently, Zhu et al. (2019) propose a video prediction model to generate new samples for improving segmentation accuracy. Moreover Hung et al. (2018) propose a self-taught learning approach based on the dense pixel-level probability maps produced by adversarial network. Mittal et al. (2019) describe the dual-branch method including a semi-supervised classification branch to learn from unlabeled data.

In this study, we propose an auxiliary network, which can determine the credibility of each pixel on the segmentation map of unlabeled images and then the trusted pixels can be used as additional

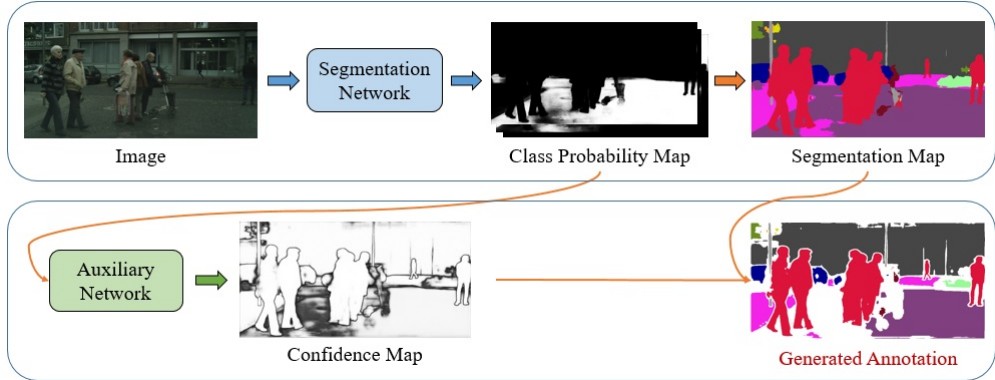

Figure 1: The process of generating annotations for unlabeled image.

ground truth data in training. Different from the previous works, we focus on the quality of the generated labels. We believe that the supervision signals of small objects are more important than the larger ones. Thus, we assign different weights to the generated tags of different categories, especially biased towards the small objects. In training the auxiliary network, we propose an auxiliary loss function containing two critical terms to achieve this objective.

## 3 METHODOLOGY

In this section, we first describe the general framework of our semi-supervised semantic segmentation scheme, and then we describe the architecture and design concepts of the proposed segmentation network and auxiliary network.

### 3.1 SEMI-SUPERVISED LEARNING USING AUXILIARY NETWORK

There are in total three training stages in our semi-supervised learning scheme. In the first stage, we simply train the segmentation network using the labeled images. In this paper, we simply use the typical cross entropy loss in the supervised training of segmentation network. Then in the second stage, an auxiliary network is trained using the proposed auxiliary loss function. Both the originally labeled images and the previously trained segmentation network are used in this stage. Our auxiliary network takes the class probability maps of size $H \times W \times C$ produced by the segmentation network as input, where $H$ and $W$ are the height and width of the image size and $C$ denotes the number of classes. The auxiliary network outputs a confidence map of size $H \times W \times 1$. Each pixel of the confidence map is a probability value between 0 and 1, which represents the credibility of that pixel on the segmentation map. In the third stage, we feed the unlabeled images into the segmentation network pre-trained in the first stage to obtain the class probability map, and then pass this map through the auxiliary network to obtain the confidence map. Based on the probability value of each pixel of the confidence map, we can determine the segmentation reliability of each pixel on the segmentation map. Then, we select the pixels with high reliability as the annotated data points. And we ignore the pixels with lower reliability in the next-phase training. We set a threshold $T_{aux}$ on the confidence map to separate the high reliability pixels from the low reliability pixels. Hence, we can mask the error-prone pixels on the output segmentation map and produce the reliable annotated maps for the unlabeled images. Finally, since all the images now have labels, we retrain the segmentation network using all of them, and produce a more accurate segmentation model. Figure 1 shows the process for generating annotated labels for the images without ground truth.

**Loss function for auxiliary network**   To train the auxiliary network, we create another auxiliary ground truth map of dimension $H \times W \times 1$. Assuming that we have the original labeled ground truth class map and the estimated (segmented) class map produced by the segmentation network, the pixel value on the auxiliary ground truth map is set to 1 if the class at that pixel on the estimated segmentation map matches that on the ground truth class map; and it is set to 0, if not. Different

from the typical binary cross entropy loss, we assign weights to the terms in the loss function for auxiliary network, which is defined below.

$$L_{aux} = -\sum_{h,w} W_{h,w}^n (\gamma_{h,w}^n \times y_{h,w}^n \log(a_{h,w}(x^n)) + (1 - y_{h,w}^n) \log(1 - a_{h,w}(x^n))) \qquad (1)$$

where $y_{h,w}^n$ equals to 1 if the estimate class on the segmentation map at pixel $(h, w)$ is identical to the class value on the ground truth label map at the same pixel, and $y_{h,w}^n$ equals to 0 if they are different. In equation (1), $a_{h,w}(x^n)$ denotes the confidence map output at pixel $(h, w)$, when the auxiliary network takes the class probability map $x^n$ as input. $W_{h,w}^n$ represents the class weighting value at pixel $(h, w)$ on the confidence map, and it is defined as $1/\ln(c + p_{class})$, inspired by ENet (Paszke et al., 2016), where we set c equal to 1.04. The class weighting at different locations of confidence map is decided by the ground truth label map. For example, when the class at the pixel $(h, w)$ on ground truth label map is class $i$, the class weight at that pixel on the confidence map is the weight value associated with class $i$. Class weighting term is used to tackle the data imbalance issue. That is, some classes having few number of pixels are often unfavored in training. With class weighting, our auxiliary network performs well not only on the big objects but also on the small objects. Moreover, for different classes, the imbalance issue between positive and negative samples are quite different. Thus, we propose a weight term $\gamma_{h,w}^n$, which is defined as $\#negative\,pixel_{class}/\#positive\,pixel_{class}$, and it is calculated using the auxiliary network ground truth. Similarly, this weight term is location variant depending on the ground truth label map. Both $W_{h,w}^n$ and $\gamma_{h,w}^n$ are the critical terms in the proposed auxiliary loss function. There were several versions tested and the current version gives the best result.

## 3.2 NETWORK ARCHITECTURE

### 3.2.1 SEGMENTATION NETWORK

We adopt the DSNet-fast (Chen et al., 2019) as our backbone model, but we made some modifications. Based on our observations, once the receptive field of the network is large enough to cover the entire image, making good use of the shallow-layer features is more effective than increasing the network depth for the semantic segmentation task. Thus, we redesigned the initial block and the early block of DSNet-fast, and reduced the CNN layers in the deeper block to achieve a better accuracy and inference speed. The overall architecture is shown in Figure 2.

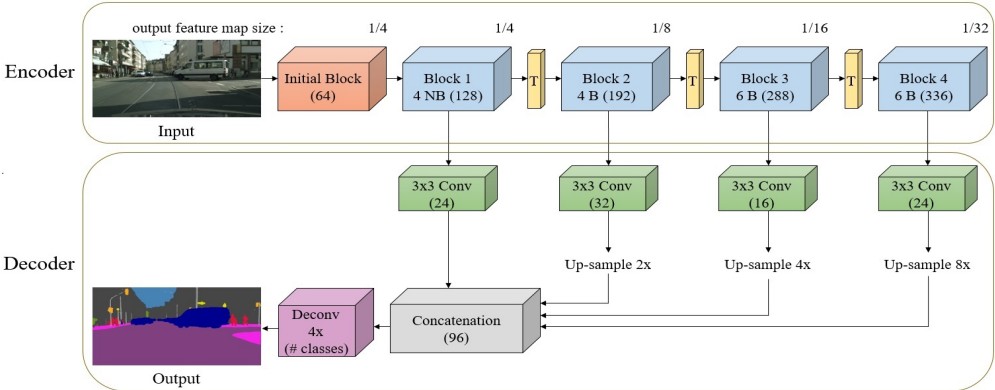

Figure 2: The architecture of the proposed segmentation network. "NB": non-bottleneck unit. "B": bottleneck unit. "T": transition layer. The number of output channels for each block are marked inside parentheses.

In the encoder part, we adopt the same bottleneck unit and transition layer in DSNet-fast, and propose new initial block and non-bottleneck units. All our core units are shown in Figure 3. In the initial block, after applying the 3×3 convolution with stride 2 on the input image, we split it into two branches. The 1×1 convolution is applied to the left branch to integrate features and reduce the number of channels, and on the right branch, a stack of two 3×3 convolutions is used to learn the

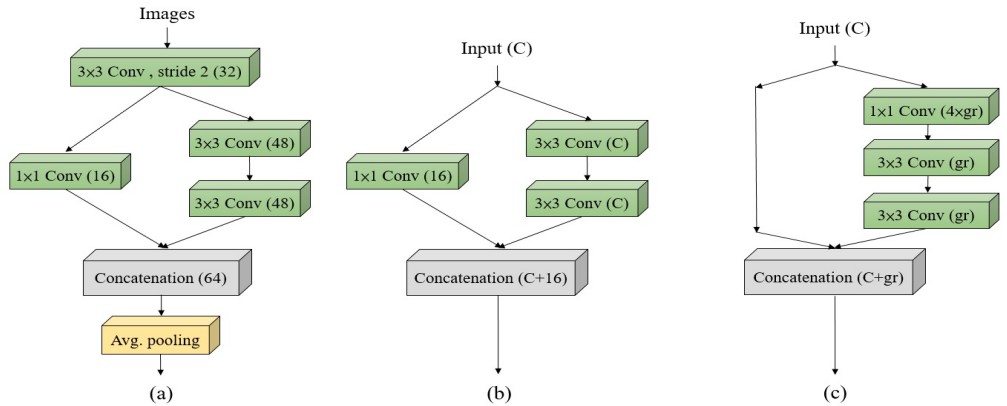

Figure 3: Proposed core units. (a) initial block. (b) non-bottleneck unit. (c) bottleneck unit. "C": the number of channels from input. "gr": growth rate, which is set to 32 in all our bottleneck unit. The number of channels for each layer are marked inside parentheses.

features. Furthermore, the feature maps of these two branches are concatenated and followed by an average pooling with stride 2 to produce the output. Since the 1×1 convolution does not take the spatial information into account, the feature map concatenated from the two branches have different receptive fields, and thus it helps the network to extract features at different scales. On the other hand, due to dimension reduction made by the 1×1 convolution operation, the left branch helps the right branch to produce more feature channels under the constraint of fixed number of concatenated feature maps. Similarly, we use the same concept to design our non-bottleneck unit. We find that this design strategy is indeed helpful in producing better segmentation results at similar model complexity.

In the decoder part, we employ 3×3 convolutions with {24, 32, 16, 24} channels after the Blocks 1-4, respectively, to reduce the computational complexity. Furthermore, the up-sampling and concatenation operations are adopted, same as DSNet-fast, to integrate the feature maps at different spatial levels. Finally, the concatenated feature map is up-sampled by a factor of 4 using a deconvolution kernel to produce the estimated segmentation map. All the convolution layers are followed by batch normalization (Ioffe & Szegedy, 2015) and ReLU activations.

### 3.2.2 AUXILIARY NETWORK

Our auxiliary network mainly consists of five convolution layers, one atrous spatial pyramid pooling (ASPP) module, and one deconvolution layer. The overall architecture is shown in Table 1. A few auxiliary network architectures have been tested before we settle on the current design, which is quite simple but provides good results.

Table 1: Auxiliary network architecture.

| Input size | Block | Output size |
|---|---|---|
| 512×1024×3 | 3×3 Conv , stride 2 | 256×512×32 |
| 256×512×32 | 3×3 Conv , stride 1 | 256×512×64 |
| 256×512×64 | 3×3 Conv , stride 2 | 128×256×128 |
| 128×256×128 | 3×3 Conv , stride 1 | 128×256×256 |
| 128×256×256 | 3×3 Conv , stride 1 | 128×256×256 |
| 128×256×256 | ASPP | 128×256×160 |
| 128×256×160 | 8×8 Deconv , 4x | 512×1024×1 |

Since we find that detailed spatial information is vital in the auxiliary network, our feature map size is only 1/4 of the original image size before the deconvolution layer. We use the $3\times3$ convolution layers at the beginning, and then it is followed by a single atrous spatial pyramid pooling module similar to DeepLab-v3+ (Chen et al., 2018) to extract multi-scale information as well as enlarging the receptive field. For the ASPP (Atrous Spatial Pyramid Pooling) module of our auxiliary network, we use the dilated convolution with dilated rate $\{1, 2, 4, 8, 16\}$ for branch 1-5 respectively, and do not use the image pooling branch. Each branch in our ASPP module has 32 feature channels. Finally, the deconvolution kernel is applied after the ASPP module to rescale the output to the input size and generate the dense confidence map; moreover, it is followed by the sigmoid function to limit the output between 0 and 1. Same as the segmentation network, all the convolution layers are followed by batch normalization (Ioffe & Szegedy, 2015) and ReLU activations.

## 4 EXPERIMENTAL RESULTS

In this section, we first introduce two datasets, Cityscapes (Cordts et al., 2016) and CamVid (Brostow et al., 2008), that are used to evaluate proposed method. Both are popular road scene datasets. Then, we describe our training details for the segmentation and the auxiliary networks. At the end, we show the comparisons with the state-of-the-art methods as well as some visual results.

### 4.1 DATASETS

**Cityscapes** The Cityscapes dataset is a road scene dataset with the image resolution of 1024 $\times$2048, and it provides 19 object classes for evaluation. There are two forms of annotation data in its database, namely, fine and coarse. Only the fine annotation set is used in our experiments, where it contains 2975, 500, 1525 images for training, validation, and testing respectively. We resize the image to $512\times1024$ for training and testing due to the hardware limitation; however, for evaluation on the testing set, we restore the resolution of segmentation results to the original size for fair comparison.

**CamVid** The CamVid dataset is also a road scene dataset, but its image resolution (about $360\times480$) is much smaller than that of the Cityscapes dataset. It provides 11 semantic classes, and contains 367, 101, 233 images for training, validation, and testing.

### 4.2 IMPLEMENTATION DETAILS

We use PyTorch framework to implement our method, and we measure the network speed on a single GTX 1080Ti GPU. The popular metric, mean IoU, is used to compute the segmentation performance. In addition, the encoder part of our segmentation network has been pre-trained on ImageNet (Deng et al., 2009) to produce better initial parameters.

**Segmentation network** To train the segmentation network for the Cityscapes dataset, we use the stochastic gradient decent (SGD) optimization with momentum 0.9, weight decay 0.0001, and batch size 4. We adopt the poly learning rate policy as described in Chen et al. (2017a) with power 0.9 in all our experiments and the initial learning rate is set to 0.05 here. We train it for 200 epochs in total. For the CamVid dataset, the SGD optimization is also used but with momentum 0.9, weight decay 0.0005, and batch size 8. In addition, the initial learning rate is set to 0.075, and we train it for 150 epochs in total.

For both Cityscapes and CamVid datasets, we adopt the class weighting scheme same as in Paszke et al. (2016) with an additional hyper-parameter c set to 1.1, and the data augmentation strategies are employed to boost the performance. Our data augmentation strategies include random horizontal flip, random pixel translation, and random scaling. For the Cityscapes dataset, the scaling factors in random scaling are $\{0.75, 1.0, 1.25, 1.5, 1.75\}$, and for the CamVid dataset, they are $\{0.8, 1.0, 1.3, 1.7\}$.

**Auxiliary network** To train the auxiliary network for Cityscapes dataset, we use the Adam optimization (Kingma & Ba, 2014) with momentum 0.9, weight decay 0.0001, batch size 4, and the

Table 2: Comparison with the state-of-the-art methods on the Cityscapes testing set. The efficiency-based methods are adopted for comparison. "†": inference speed using Titan X GPU. "‡": inference speed using Titan XP GPU. "††": inference speed using GTX 1080Ti GPU.

| Method | Training dataset | Mean IoU (%) | Speed (FPS) | Params |
|---|---|---|---|---|
| ENet (Paszke et al., 2016) | Fine | 58.3 | 76.9† | 0.37 M |
| ContextNet (Poudel et al., 2018) | Fine | 66.1 | 18.3† | 0.85 M |
| EDANet (Lo et al., 2018) | Fine | 67.3 | 81.3† | 0.68 M |
| Fast-SCNN (Poudel et al., 2019) | Fine, Coarse | 68.0 | 75.3† | 1.11 M |
| BiSeNet (Yu et al., 2018) | Fine | 68.4 | 105.8‡ | 5.8 M |
| DSNet-fast (Chen et al., 2019) | Fine | 69.1 | 52.9† | 3.0 M |
| ICNet (Zhao et al., 2018) | Fine | 69.5 | 30.3† | 6.68 M |
| ERFNet (Romera et al., 2017) | Fine | 69.7 | 41.7† | 2.1 M |
| DF1-Seg-d8 (Li et al., 2019) | Fine | 71.4 | 136.9†† | - |
| DF2-Seg2 (Li et al., 2019) | Fine | 75.3 | 56.3†† | - |
| Ours | Fine | 73.9 | 55.8† , 73.2†† | 2.11 M |

initial learning rate is set to 0.0005. For the CamVid dataset, different from the former, the weight decay for Adam optimization is set to 0.0005, and batch size is set to 8.

The training duration is 50 epochs for both datasets, and we use the proposed auxiliary loss function in training the auxiliary network. The data augmentation strategies for training auxiliary network are same as described in the above.

### 4.3 PERFORMANCE EVALUATION

**Evaluation on Cityscapes dataset**   In order to evaluate the performance of the proposed segmentation network, we test it on the Cityscapes testing set, and compare the results with the state-of-the-art networks. It is worth noticing that we do not adopt any testing technology such as multi-scale testing in the evaluation process. From Table 2, we find that our segmentation network can achieve a good trade-off between speed and accuracy. It achieves 73.9% mean IoU with fewer parameters and higher inference speed comparing to the other segmentation nets.

Moreover, to evaluate the proposed semi-supervised learning scheme, similar to that in work (Hung et al., 2018), we randomly sample half of images from the training set and call them the labeled data, and the other half are the unlabeled data. We conduct the experiments on the Cityscapes validation set. First of all, we train the segmentation and auxiliary network using the labeled images as described in Section 3.1, and then we measure the confidence map performance. In this process, we feed the validation set images to the entire network, and then mask the error-prone pixels of the segmentation map by the confidence map. Then, compare the masked segmentation map (generated annotations) with its true ground truth to calculate the accuracy. We use three performance metrics for evaluation, and the experimental results are shown in Table 3, where $T_{aux}$ represents the threshold set for the confidence map. The mean IoU metric is the mIoU calculated by the generated annotations and the true ground truth without considering the masked pixels (by the confidence map). The *average class semi ratio* is the average value of the "class semi ratio" of all classes. To compute the *class semi ratio* for class $i$, we only consider the region of pixels that belong to class $i$ on the ground truth label map. The denominator of class semi ratio for class $i$ is the number of all pixels; and the numerator is the number of pixels not masked by the confidence map, on the estimated segmentation map. In calculating the *selected pixels*, we first count the pixels of the estimated segmentation map that are selected by the confidence map and are used as ground truth (GT) in training. Then, we compute its ratio with respect to the entire image. From Table 3, we can find that with a higher threshold, the pixels selected as GT are more accurate, but the selected pixels are

Table 3: The measuring results for generated annotations on the Cityscapes validation set using half of training data.

| $T_{aux}$ | Mean IoU (%) | Average class semi ratio (%) | Selected pixels (%) |
|---|---|---|---|
| 0 | 70.3 | 100 | 100 |
| 0.7 | 83.7 | 75.6 | 86.1 |
| 0.8 | 85.6 | 71.2 | 83.3 |
| 0.9 | 88.2 | 64.2 | 78.0 |

Table 4: The per-class IoU for generated annotations on the Cityscapes validation set using half of training data.

| Class | IoU (%) / Class semi ratio (%) | | | |
|---|---|---|---|---|
| | $T_{aux} = 0.0$ | $T_{aux} = 0.7$ | $T_{aux} = 0.8$ | $T_{aux} = 0.9$ |
| Road | 97.4 / 100 | 99.0 / 95.7 | 99.2 / 95.0 | 99.4 / 93.4 |
| Sidewalk | 80.5 / 100 | 91.0 / 83.0 | 92.1 / 79.9 | 93.5 / 75.0 |
| Building | 90.9 / 100 | 96.5 / 85.4 | 97.1 / 82.5 | 97.8 / 77.4 |
| Wall | 49.2 / 100 | 62.1 / 67.9 | 63.8 / 62.8 | 66.7 / 54.6 |
| Fence | 52.8 / 100 | 67.5 / 68.5 | 70.0 / 63.0 | 73.8 / 54.8 |
| Pole | 59.4 / 100 | 78.0 / 72.2 | 82.0 / 66.3 | 87.5 / 56.0 |
| Traffic light | 63.5 / 100 | 81.6 / 74.4 | 85.2 / 68.7 | 89.9 / 59.3 |
| Traffic sign | 71.9 / 100 | 88.0 / 79.5 | 90.3 / 75.1 | 93.2 / 67.8 |
| Vegetation | 91.3 / 100 | 97.4 / 86.1 | 97.9 / 83.4 | 98.6 / 78.7 |
| Terrain | 59.4 / 100 | 74.1 / 72.2 | 76.3 / 67.7 | 80.1 / 60.4 |
| Sky | 93.5 / 100 | 98.6 / 91.4 | 98.8 / 89.5 | 99.2 / 86.4 |
| Person | 75.5 / 100 | 90.6 / 80.5 | 92.8 / 75.8 | 95.2 / 67.6 |
| Rider | 53.8 / 100 | 71.1 / 67.2 | 75.3 / 59.0 | 81.0 / 46.8 |
| Car | 92.7 / 100 | 98.6 / 89.4 | 98.9 / 87.4 | 99.3 / 83.9 |
| Truck | 68.7 / 100 | 86.7 / 70.8 | 88.0 / 67.4 | 89.9 / 62.3 |
| Bus | 70.9 / 100 | 89.1 / 70.6 | 90.8 / 67.3 | 93.1 / 62.2 |
| Train | 51.3 / 100 | 71.0 / 52.9 | 73.1 / 47.4 | 76.0 / 39.9 |
| Motorcycle | 42.7 / 100 | 62.3 / 52.4 | 65.1 / 44.5 | 68.3 / 33.3 |
| Bicycle | 70.1 / 100 | 86.7 / 76.8 | 89.9 / 70.7 | 93.6 / 60.6 |

fewer; this is a trade-off. In addition, as shown in Table 4, our confidence map can also produce good results on the small objects with a higher threshold. Thus, the generated annotations used for retraining segmentation network have good quality.

After generating the annotated labels for 1/2 unlabeled images, we retrain the segmentation network using all images. The experimental results of selecting pixels based on different threshold values are shown in Table 5. The proposed semi-supervised learning scheme can improve the performance for about 1.0% to 1.2%. If $T_{aux}$ is set to 0, it means that all the pixels of the estimated segmentation map are used as ground truth for unlabeled image, which is obviously unreasonable and it decreases

Table 5: The proposed semi-supervised learning segmentation results on the Cityscapes validation set based on different threshold values on the confidence map.

| Data amount | | $T_{aux}$ | Mean IoU (%) |
|---|---|---|---|
| Labeled data | Unlabeled data | | |
| 1/2 | 1/2 | 0 | 69.9 |
| 1/2 | 1/2 | 0.7 | 71.3 |
| 1/2 | 1/2 | 0.8 | 71.4 |
| 1/2 | 1/2 | 0.9 | 71.5 |
| 1/2 | 1/2 | 1 | 70.3 |

Table 6: Comparison with Hung et al. (2018) and Mittal et al. (2019) on the Cityscapes validation set.

| Method | Data amount (labeled data + unlabeled data) | | | |
| --- | --- | --- | --- | --- |
| | 1/8 + 7/8 | 1/4 + 3/4 | 1/2 + 1/2 | 1 + 0 |
| (Hung et al., 2018) baseline | 55.5 | 59.9 | 64.1 | 66.4 |
| (Hung et al., 2018) + adversarial learn. | 57.1 | 61.8 | 64.6 | 67.7 |
| (Hung et al., 2018) + adversarial learn. + semi | 58.8 | 62.3 | 65.7 | - |
| (Mittal et al., 2019) baseline | 56.2 | 60.2 | - | 66.0 |
| (Mittal et al., 2019) + adversarial learn. + semi | 59.3 | 61.9 | - | 65.8 |
| Ours baseline | 56.8 | 63.0 | 66.1 | 69.6 |
| Ours + semi | 60.7 | 65.5 | 67.7 | - |
| Ours baseline (using proposed segment. net) | 57.7 | 65.5 | 70.3 | 74.4 |
| Ours + semi (using proposed segment. net) | 60.2 | 68.4 | 71.5 | - |

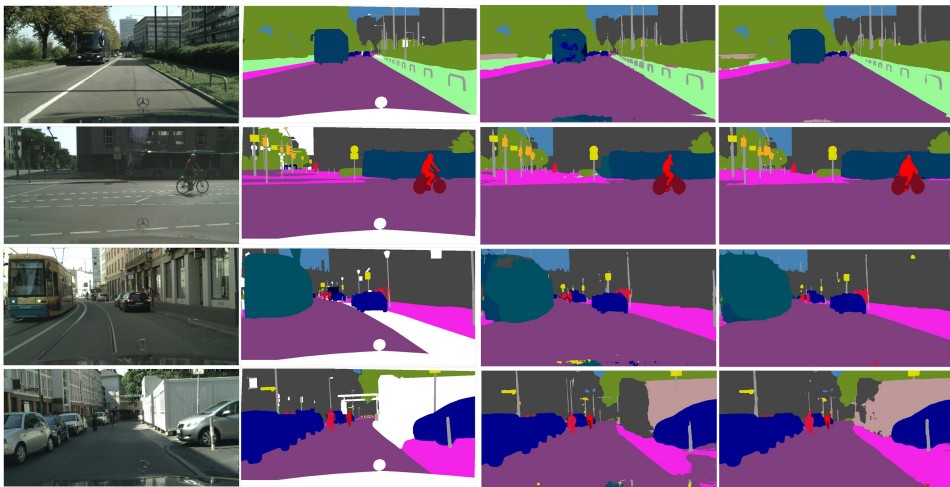

Figure 4: Sample results on the Cityscapes validation set using half of training data. From left to right: (a) image (b) ground truth (c) segmentation results without semi-supervised learning (d) segmentation results with semi-supervised learning

the accuracy. Since $T_{aux}$ at 0.9 gives the best results, we use the same value in the following experiments.

To verify the effectiveness of our approach, we conduct experiments with different amount of data, and shows the comparison with Hung et al. (2018) and Mittal et al. (2019) in Table 6. For the fair comparison, we also adopt the DeepLab-v2 framework as our backbone segmentation net model. Under the same settings that using 1/2, 1/4, and 1/8 training images as labeled data and using the rest of images as unlabeled data, our semi-supervised learning method can lead to higher performance improvements. It can improve the mean IoU by 3.9%, 2.5%, and 1.6% for 1/8, 1/4, and 1/2 training datasets, respectively. Particularly, compared with Hung et al. (2018), if we focus on the performance differences between the schemes with and without semi-supervised learning mechanism, our semi-supervised learning approach offers quite significant improvement. Furthermore, when our proposed segmentation network is used as the backbone segmentation net, it increases the mean IoU for another 1.2% to 2.9%. At the end, we show some sample visual results in Figure 4.

**Evaluation on CamVid dataset** We also evaluate our scheme on the CamVid dataset. As the results shown in Table 7, our semi-supervised learning method can boost the performance of the segmentation network, and it can achieve 1.3% improvement on mean IoU.

Table 7: Comparisons on the CamVid testing set.

| Method | Data amount | | Mean IoU (%) |
|---|---|---|---|
| | Labeled data | Unlabeled data | |
| Ours baseline (using proposed segment. net) | 1 | 0 | 71.8 |
| Ours baseline (using proposed segment. net) | 1/2 | 1/2 | 69.3 |
| Ours + semi (using proposed segment. net) | 1/2 | 1/2 | 70.6 |

## 5 CONCLUSIONS

In this paper, we first propose a highly efficient segmentation network as our platform, and then we design a semi-supervised learning scheme using an auxiliary network. The auxiliary network is used to verify the estimated segmentation map and to generate annotations on the unlabeled images. Equipped with the carefully designed auxiliary loss function in training, the auxiliary network performs well not only on the large objects but also on the small objects. It shows that the unlabeled images together with the generated annotations (labels) can be used to retrain the segmentation network for better segmentation quality. Our experimental results on the Cityscapes and CamVid datasets demonstrate the effectiveness of the proposed method.

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
