# OpenReview forum: "Semi-supervised Semantic Segmentation using Auxiliary Network"
_ICLR.cc/2020/Conference — Reject_

### Official Review · AnonReviewer1 · 2019-10-17
**Official Blind Review #1**

**Rating:** 3

**Review:**

This submission introduces a semi-supervised method using auxiliary network for improved semantic segmentation. The authors modify a previous work as their main network architecture and use another small network as auxiliary branch. The framework can work in a semi-supervised setting since they can use confidence map to annotate unlabeled images to train the network.

I give an initial rating of weak reject because (1) novelty in architecture design is trivial (2) the way of using unlabeled images is not new (3) experiments are not supportive (3) performance is not comparative to state-of-the-art. I will illustrate more as below.

1. Architecture is not novel. As I mentioned, the authors adopt DSNet-fast as their main branch with minor modifications. And they use a simplified DeepLab architecture as their auxiliary branch. There is no ablation study or strong motivation to design this network.

2. Using an auxiliary branch to measure confidence and try to get more labeled data is not new. There are many previous literature exploring similar ideas. Please clarify the differences between the literature and this submission in either introduction or related work section.
    (1) Universal Semi-Supervised Semantic Segmentation, ICCV 2019
    (2) Improving Semantic Segmentation via Video Propagation and Label Relaxation, CVPR 2019
    (3) Semi-Supervised Semantic Segmentation with High- and Low-level Consistency, Arxiv 2019
Especially for (2) and (3), (2) use unlabeled images with confidence map, (3) has two branches as well. Both of them has results on Cityscapes.

3. In table2 , speed comparison is not intuitive. The methods are evaluated on too many kinds of hardware and can not be directly compared. It is good to re-evaluate all algorithms on a single hardware, or remove this column.

4. The experiment setting is too simple. Divide existing training data into half will make it hard to compare with other approaches. All the experiments in this submission can only show you are better than the baselines, but can't convince me your approach actually works. Because the training distribution is similar, it is easy for the auxiliary network to generalize. I would suggest the authors to use all the training data, and bring additional unlabeled images into picture to see what will happen. In addition, if you use all training data, you can make fair comparisons to many literatures and demonstrate the effectiveness of your approach.

5. What is "ours" and "ours with proposed network" in table 5, please clarify.

**Experience Assessment:**

I have published one or two papers in this area.

**Review Assessment: Checking Correctness Of Derivations And Theory:**

I carefully checked the derivations and theory.

**Review Assessment: Checking Correctness Of Experiments:**

I carefully checked the experiments.

**Review Assessment: Thoroughness In Paper Reading:**

I read the paper thoroughly.

---

### Official Review · AnonReviewer2 · 2019-10-23
**Official Blind Review #2**

**Rating:** 3

**Review:**

- This paper proposes a semi-supervised learning strategy for semantic segmentation of road scenes. Specifically, authors propose to include an auxiliary network that will predict the confidence (at pixel-level) of the predictions on unlabeled images. These confidence values will be used to  generate a new auxiliary ground-truth to retrain the network using the unlabeled images.
- Even though the idea is somehow interesting and results seem to improve with respect to the baselines, this paper is very similar to standard semi-supervised learning approaches for natural images that employ image proposals (i.e., EM-based methods). In those works, unlabeled images are segmented with the network trained on labeled images, generating some proposals. These proposals are later employed as a fake ground truth to re-train the network employing both labeled and unlabeled images. The only difference in this work is to employ the virtual confidence map to mask-out some pixels (those with lowest confidence values).
- Related work section is extremely weak. Authors merely mention few papers (some other relevant papers are missing), and throw a sentence for each one, without making connections between works. This makes difficult to place their work among the literature (e.g., which limitations of previous approaches the current method intend to address?). Authors should significantly improve this section.
- I am not sure about the fact that employing only the probability maps as input to generate a confidence map is reliable, if no other information is employed. These predictions (e.g., confidence map) will be based only on the probabilities obtained by the first network. While this is already a good indicator of the confidence of the network to make those predictions, I believe that input images should also be included. The intuition behind this is that there may exist some regions with similar probabilities (from first network), which are incorrectly classified (leading to 0-masked pixels on the auxiliary ground-truth) in some cases, while correctly classified in other situations (leading to 1-masked pixels on the auxiliary ground-truth).
- Eq.1) is basically the standard cross-entropy loss, with the difference of the weighting terms.
- a_{h,w} is the softmax of the auxiliary network, isn’t it?
- Further, authors threshold the values of the confidence map to generate the new auxiliary ground truth. Why not to use the raw values so that each pixel is weighted differently according to its importance?
- Authors make some claims which were never demonstrated. For example, they mention that the proposed approach performs better on small targets than previous approaches. Nevertheless, only mean results (over all the classes) are shown. To this end authors should report per-class performances, instead of the mean.
- Furthermore, authors make several over claims, misleading information. For example, they mentioned that they proposed a highly efficient segmentation method. Nevertheless, from Table 2 it can be observed that the proposed method ranks in the middle in terms of both speed and parameters, compared to other state-of-the-art models. Similarly, authors mention that their model is equipped with a carefully designed auxiliary loss function during training, while they basically employ a standard cross-entropy weighted by some values to account for imbalance between classes and between positive and negative pixels within the same class.
- The paper contains many grammatical errors.



**Experience Assessment:**

I have published in this field for several years.

**Review Assessment: Checking Correctness Of Derivations And Theory:**

I carefully checked the derivations and theory.

**Review Assessment: Checking Correctness Of Experiments:**

I carefully checked the experiments.

**Review Assessment: Thoroughness In Paper Reading:**

I read the paper thoroughly.

---

### Official Review · AnonReviewer3 · 2019-10-24
**Official Blind Review #3**

**Rating:** 3

**Review:**

This paper focused on the problem of semantic segmentation. The author first proposed an efficient segmentation network. Then a semi-supervised learning scheme with an auxiliary network is introduced to annotate the unlabeled images thus help boost the segmentation accuracy.

Clarity:
There is no novelty for the proposed fast segmentation network detailed in Sec 3.2. The auxiliary network for predicting the confidence map may look interesting, while the experimental results are not convincing.

Experiments:
1. The design of the fast segmentation network in this paper is boring and not much related to the title.

2. The experimental results regarding the effectiveness of the auxiliary network are based on a weak backbone, some stronger backbones like DeepLab and PSPNet should be included.

3. What the results would be if using all labeled data and the newly added unlabeled data? Can the newly unlabeled data with labels from the auxiliary network help boosting the performance?


**Experience Assessment:**

I have published in this field for several years.

**Review Assessment: Checking Correctness Of Derivations And Theory:**

I carefully checked the derivations and theory.

**Review Assessment: Checking Correctness Of Experiments:**

I carefully checked the experiments.

**Review Assessment: Thoroughness In Paper Reading:**

I read the paper thoroughly.

---

### Decision · Program_Chairs · 2019-12-19

**Decision:**

Reject

**Comment:**

The paper presents a semi-supervised learning approach to handle semantic classification (pixel-level classification). The approach extends Hung et al. 18, using a confidence map generated by an auxiliary network, aimed to improve the identification of small objects.

The reviews state that the paper novelty is limited compared to the state of the art; the reviewers made several suggestions to improve the processing pipeline (including all images, including the confidence weights).
The reviews also state that the paper needs be carefully polished.

The area chair hopes that the suggestions about the contents and writing of the paper will help to prepare an improved version of the paper.